# Revenge of the Fallen? Recurrent Models Match Transformers at Predicting Human Language Comprehension Metrics

**James A. Michaelov**[a]    **Catherine Arnett**[b]    **Benjamin K. Bergen**[a]
[a]Department of Cognitive Science, [b]Department of Linguistics,
University of California San Diego
{j1michae, ccarnett, bkbergen}@ucsd.edu

## Abstract

Transformers have generally supplanted recurrent neural networks as the dominant architecture for both natural language processing tasks and for modelling the effect of predictability on online human language comprehension. However, two recently developed recurrent model architectures, RWKV and Mamba, appear to perform natural language tasks comparably to or better than transformers of equivalent scale. In this paper, we show that contemporary recurrent models are now also able to match—and in some cases, exceed—performance of comparably sized transformers at modeling online human language comprehension. This suggests that transformer language models are not uniquely suited to this task, and opens up new directions for debates about the extent to which architectural features of language models make them better or worse models of human language comprehension.

## 1    Introduction

The origins of recurrent neural networks lie in attempts to model human cognition, and specifically the human language system (Jordan, 1986; Elman, 1990). Following improvements such as long short-term memory (LSTM; Hochreiter & Schmidhuber, 1997; Gers et al., 2000), recurrent neural networks were for a while the dominant architecture not only for modeling human language comprehension (e.g. Frank et al., 2015), but for natural language systems in general (see, e.g. Goldberg, 2016). In recent years, they have in turn been superseded by transformer language models, which empirically tend to show better performance at both a range of natural language tasks (see, e.g., Radford et al., 2019; Dai et al., 2019) and at predicting metrics of human language comprehension (e.g. Wilcox et al., 2020; Merkx & Frank, 2021; Michaelov et al., 2022). Nonetheless, the question of how recurrent and transformer language models compare as cognitive models of the human language system is still an open one. One the one hand, recurrent neural networks inherently model the process of maintaining a specific informational state and integrating this with new information as it occurs incrementally. This principle is widely believed to underlie language comprehension and other real-time processing (Merkx & Frank, 2021; Michaelov et al., 2021). On the other hand, transformers have been argued to better model cue-based retrieval accounts of language comprehension (Ryu & Lewis, 2021; Merkx & Frank, 2021), and their direct access to previous words may allow them to better model human-like lexical priming effects (Michaelov et al., 2021). In addition, transformers' superior performance at predicting metrics of human language comprehension in itself serves as evidence that, at the very least, the statistical patterns learned by transformer language models capture something also learned by humans.

As they have increased in scale (number of parameters, number of training tokens, or both), transformers have been found to improve at natural language tasks (Brown et al., 2020; Kaplan et al., 2020; Rae et al., 2022; Hoffmann et al., 2022; Chowdhery et al., 2022; Touvron et al., 2023), as well as at predicting both behavioral (Wilcox et al., 2020; Merkx & Frank, 2021) and neural (Merkx & Frank, 2021; Michaelov et al., 2022) metrics of online language comprehension. But in recent years, two wrinkles have emerged. The first is

evidence that larger models and those trained on more data may actually predict some behavioral metrics of language comprehension (such as reading time) worse than smaller models do (Kuribayashi et al., 2021; Oh et al., 2022; Oh & Schuler, 2023a;b; Oh et al., 2024; Shain et al., 2024). Second, two recently developed recurrent language model architectures appear to perform natural language tasks at least as well as transformers of equivalent size and training: RWKV (Peng et al., 2023) and Mamba (Gu & Dao, 2023). Transformers are therefore no longer the definitively best-performing language model architecture, and it is no longer the case that we should expect further advances in transformers to necessarily lead to improved fit to metrics of human language comprehension. Thus the time is ripe to revisit the question of which language model architecture best predicts human language comprehension.

To this end, we compare the performance of the Pythia (Biderman et al., 2023), RWKV (Peng et al., 2023), and Mamba (Gu & Dao, 2023) suites of autoregressive language models on 12 human language comprehension datasets (Federmeier et al., 2007; Hubbard et al., 2019; Michaelov et al., 2024; Szewczyk & Federmeier, 2022; Szewczyk et al., 2022; Wlotko & Federmeier, 2012; Boyce & Levy, 2023; Brothers & Kuperberg, 2021; Futrell et al., 2021; Kennedy et al., 2003; Luke & Christianson, 2018; Smith & Levy, 2013) covering 5 different metrics. Since all models were trained on the same dataset and have comparable numbers of parameters, we are able to measure the effect of architecture on the extent to which a language model's predictions correlate with metrics of human language comprehension.

## 2 Modeling prediction in human language comprehension

Over the years, a wide range of language models have been used to model data from experiments on human language comprehension, including **n-gram models** (e.g., McDonald & Shillcock, 2003; Boston et al., 2008; Demberg & Keller, 2008; Mitchell et al., 2010; Smith & Levy, 2013; Brothers & Kuperberg, 2021), **recurrent neural networks** (**RNNs**; e.g., Frank & Bod, 2011; Fossum & Levy, 2012; Monsalve et al., 2012; Frank et al., 2015; Goodkind & Bicknell, 2018; Aurnhammer & Frank, 2019), and most recently, **transformers** (e.g., Wilcox et al., 2020; Hao et al., 2020; Merkx & Frank, 2021; Kuribayashi et al., 2021; Szewczyk & Federmeier, 2022; Michaelov et al., 2022; 2024; Boyce & Levy, 2023; Wilcox et al., 2023a;b; Oh et al., 2022; 2024; Oh & Schuler, 2023a;b; Shain et al., 2024). Each approach can be evaluated in terms of how well it performs either as a computational-level model (in the vein of Marr, 1982) or as a cognitive model. Language models can serve as computational-level models since they calculate the probability of a word in a given context; and thus, their predictions can be compared with analogous measures of prediction in humans. Humans may also be able to predict the probability of words based on the statistics of language, given evidence that they are sensitive to statistical properties of language such as word frequency (Van Petten & Kutas, 1990; Van Petten, 1993; Dambacher et al., 2006; Rugg, 1990; Fischer-Baum et al., 2014; Shain, 2024). This opens a door for thinking of language models as plausible cognitive models, something further supported by recent work arguing that language models display linguistic competence (Piantadosi, 2023; Mahowald et al., 2024).

Existing studies vary in where they fall along the computational-to-cognitive model continuum. At the computational end of the scale, a range of studies (Smith & Levy, 2013; Brothers & Kuperberg, 2021; Meister et al., 2021; Wilcox et al., 2023b; Hoover et al., 2023; Shain et al., 2024; Michaelov & Bergen, 2022; 2024) have focused on understanding the mathematical relationship between language model probability and processing difficulty, which does not necessarily require considering the specific model features. On the other end of the scale, several researchers have argued that contemporary transformers have a strong structural resemblance to the human language system (Schrimpf et al., 2021; Hosseini et al., 2024).

Most studies fall somewhere between the two extremes. This is particularly evidenced in work on the N400, a neural signal that is often considered to index the extent to which a word has been predicted based on its preceding context (DeLong et al., 2005; Van Petten & Luka, 2012; DeLong et al., 2014; Kuperberg et al., 2020). Frank & Willems (2017), for example, explicitly choose to use a model with a modified n-gram architecture to investigate the role that pure word-level surface-level statistics may have on language comprehension. More

recently, Michaelov et al. (2024) used GPT-3 to investigate the extent to which prediction based on language statistics alone could account for several known N400 effects, including the fact that more semantically plausible words are processed more easily. We can also use this approach to test the viability of specific hypotheses about the language comprehension system by comparing language models that instantiate different theories of language comprehension. Frank et al. (2015), for example, investigate how well a traditional recurrent neural network predicts N400 amplitude compared to a model implementing a probabilistic phrase-structure grammar. They find that the former out-performs the latter, which they argue suggests that prediction during human language comprehension may rely more on statistical properties of language than on explicit hierarchical grammatical structure.

In the present study, we focus on the difference between recurrent language models and transformers. Many accounts theorize that human language comprehension involves construction of lossy and in some cases incorrect representations of the utterance and its meaning, due to cognitive resource limitations (Ferreira, 2003; Christiansen & Chater, 2016; Futrell et al., 2020), and it has been argued that recurrent models are a good computational models of this (Merkx & Frank, 2021; Michaelov et al., 2021). This is perhaps most clear in the case of the *now-or-never bottleneck* account of language comprehension, under which our limited working memory means that 'the brain must compress and recode linguistic input as rapidly as possible' (Christiansen & Chater, 2016). In this case, the analogy with recurrent models is clear—a core component of such models is that they can take inputs of any length, but are limited in that any context must be compressed into a representation with a fixed size, which is updated with each new word.

This is not the case for transformers, however. While they have fixed (though ever-increasing, see, e.g., Llama Team, 2024) context windows, they have perfect access to all the representations of words within these context windows. Thus, their representations of a word's context are not incrementally-compressed versions of the input like recurrent models; but rather, their representations of the context expand with each new word—within their context window, they have 'unlimited working memory' (Merkx & Frank, 2021), which puts them at odds with accounts such as the now-or-never bottleneck theory of language comprehension. On the other hand, such seemingly lossless working memory may be more human-like than it may first appear. As Michaelov et al. (2021) note, humans do maintain specific past words in working memory, and indeed, there is evidence that reading a given word can lead to that word being easier to process for up to 45 minutes in some specific contexts (Besson et al., 1992; for discussion see Rommers & Federmeier, 2018). Furthermore, transformers have been argued to provide a good computational-level model of cue-based retrieval accounts of language comprehension (Ryu & Lewis, 2021; Merkx & Frank, 2021). Specifically, under cue-based retrieval accounts (McElree et al., 2003; Van Dyke & Lewis, 2003; for review see Lewis et al., 2006; Parker et al., 2017), words are retrieved during language comprehension based on the features of previous words in the context which are used as *cues*; and analogously, it has been argued, the features of the representations of the words that transformers attend to when they predict the next word can be considered to function as cues to which word should be predicted (Ryu & Lewis, 2021; Merkx & Frank, 2021).

Beyond *a priori* cognitive plausibility, empirical studies with N400 data have almost universally shown that transformers out-perform recurrent neural networks, and larger transformers trained on more data (and with lower perplexities) generally perform best at predicting N400 amplitude (Merkx & Frank, 2021; Michaelov et al., 2022; Michaelov & Bergen, 2022).

Language models have also been used to model reading time, which has also been hypothesized to reflect prediction in language comprehension. However, in this area, the results have been less straightforward. Smaller models display the same pattern seen with the N400, where larger language models trained on more data and with lower perplexities perform better (Goodkind & Bicknell, 2018; Merkx & Frank, 2021; Wilcox et al., 2020; 2023a; Hao et al., 2020). But past a certain number of parameters or training tokens, their performance appears to deteriorate (Kuribayashi et al., 2021; Oh et al., 2022; 2024; Oh & Schuler, 2023a;b; Shain et al., 2024). On the question of whether recurrent neural networks or transformers best predict reading time, the results have been mixed (Wilcox et al., 2020; Eisape et al., 2020; Kuribayashi et al., 2021), and have been found to differ depending on which metric of reading time is investigated (Merkx & Frank, 2021).

| RWKV-4 | | Pythia | | Mamba | |
|---|---|---|---|---|---|
| **Name** | **Parameters** | **Name** | **Parameters** | **Name** | **Parameters** |
| 169M | 169,342,464 | 160M | 162,322,944 | 130M | 129,135,360 |
| 430M | 430,397,440 | 410M | 405,334,016 | 370M | 371,516,416 |
| - | - | 1B | 1,011,781,632 | 790M | 793,204,224 |
| 1.5B | 1,515,106,304 | 1.4B | 1,414,647,808 | 1.4B | 1,372,178,432 |
| 3B | 2,984,627,200 | 2.8B | 2,775,208,960 | 2.8B | 2,768,345,600 |

Table 1: All the models used in our analysis, displaying the model's named size and the size as calculated using PyTorch. Models of comparable size are displayed next to each other. Further details for each model are provided in the cited papers and their linked repositories.

The advent of new recurrent architectures that are increasingly feasible to train at a large scale and that can perform as well as or better than transformers—namely, RWKV and Mamba— is thus important in two ways. First, it allows us to test whether the patterns previously observed in transformers—that larger and better models predict N400 amplitude better but past a certain point predict reading time worse—also holds for other architectures with comparable natural language processing performance. Second, and perhaps more crucially, it allows us to again evaluate whether, when matched on scale or performance, recurrent or transformer architectures are better models of online human language comprehension.

## 3 Method

### 3.1 Language Model Architectures

The aim of this study is to investigate how well metrics of online human language comprehension can be predicted using three types of language model: the Pythia suite of autoregressive transformers (Biderman et al., 2023); and the recurrent RWKV (Peng et al., 2023) and Mamba models (Gu & Dao, 2023). All models are trained on the Pile, a 300B token English-language dataset (Gao et al., 2020). For each architecture, we selected models of comparable size (i.e., weight class) as shown in Table 1. We discuss each architecture below.

**Pythia** Pythia (Biderman et al., 2023) is a set of autoregressive transformer models trained to be comparable across different model sizes, ranging from 70M to 12B parameters. The architecture and hyperparameters are based on GPT-3 (Brown et al., 2020), with the addition of some changes based on recent advancements (Dao et al., 2022; Su et al., 2024; Wang & Komatsuzaki, 2021; Belrose et al., 2023).

**RWKV** RWKV is a language model architecture described by its creators as a 'Reinvent[ion of the] RNN for the Transformer Era' (Peng et al., 2023). RWKV models combine the parallelizable training of transformers with unlimited context lengths, as well as several additional features that make them RNN-like. First, their time-mixing block—which can mathematically formulated in a similar way to the recurrent states of an RNN (Peng et al., 2023)—allows the representations of past states to be combined with those of new words. In addition, RWKV models explicitly have a decay parameter such that tokens earlier in the context will be weighted less than later tokens during inference, thereby explicitly introducing something analogous to working memory limitations (Merkx & Frank, 2021).

**Mamba** Mamba is another recent recurrent model architecture (Gu & Dao, 2023). One of the key goals of the Mamba architecture is to allow models to optimally compress their contexts, and especially very long contexts, into a state of fixed size such that they are still able to predict effectively. Like RWKV, Mamba computational complexity scales linearly with sequence length while avoiding the quadratic complexity of transformers (Gu & Dao, 2023). This is achieved by using a novel 'selective scan' mechanism that filters the input to select the most important information. Thus, Mamba models intuitively function like the

| Dataset | Metric | Stimuli | N | Trials |
|---|---|---|---|---|
| Federmeier et al. (2007) | N400 | 564 | 32 | 7,856 |
| Hubbard et al. (2019) | N400 | 192 | 32 | 5,705 |
| Michaelov et al. (2024) | N400 | 500 | 50 | 5,526 |
| Szewczyk & Federmeier (2022) | N400 | 600 | 26 | 4,822 |
| Szewczyk et al. (2022) | N400 | 672 | 32 | 4,939 |
| Wlotko & Federmeier (2012) | N400 | 300 | 16 | 4,440 |
| Boyce & Levy (2023) | Maze Response Time | 9,304 | 63 | 56,447 |
| Brothers & Kuperberg (2021) | SPR Three-Word RT | 648 | 216 | 46,092 |
| Futrell et al. (2021) | SPR Response Time | 9,303 | 181 | 1,566,641 |
| Kennedy et al. (2003) | Go-Past Duration | 38,186 | 10 | 195,507 |
| Luke & Christianson (2018) | Go-Past Duration | 2,399 | 84 | 105,570 |
| Smith & Levy (2013) | SPR Response Time | 6,297 | 35 | 119,120 |

Table 2: A description of each of the datasets, including the metric, the number of stimuli, the number of experimental participants (*N*), and the number of trials. See §3.2 for a brief explanation of each metric, and Appendix A for further details of each dataset.

more recent recurrent neural network variants—crucially, they include a latent state that is updated with each new input (like recurrent layers), and their selective scan method filters input (much like gating mechanisms in gated recurrent units or long short-term memory).

## 3.2 Datasets

In this study, we use language models of each of the three architectures discussed in §3.1 to model 5 metrics of human language processing from 12 datasets, the details of which are given in Table 2. These datasets comprise 6 **N400** datasets (Federmeier et al., 2007; Hubbard et al., 2019; Michaelov et al., 2024; Szewczyk & Federmeier, 2022; Szewczyk et al., 2022; Wlotko & Federmeier, 2012) and 6 reading time datasets. The latter comprise four types of reading time metric: the interval between when a word is first fixated by a reader and when they first move onto the next word, as calculated using eye-tracking (**Go-Past Duration** or **GPD**; Kennedy et al., 2003; Luke & Christianson, 2018), the time taken to click to move onto the next word in a self-paced reading task (**Self-Paced Reading Response Time** or **SPR RT**; Futrell et al., 2021; Smith & Levy, 2013), the total Response Time for a word and the two following words (to account for spillover effects) in a self-paced reading task (**Self-Paced Reading Three-Word Response Time** or **3W-RT**; Brothers & Kuperberg, 2021), and the time taken to respond to each word on the Maze task (**Maze Response Time** or **Maze RT**; Boyce & Levy, 2023). Further details of each metric and dataset are provided in Appendix A.

## 3.3 Evaluation Procedure

We used the language models discussed in §3.1 to calculate the surprisal of all critical words in all datasets given their context. For the N400 and the Brothers & Kuperberg (2021) datasets, this context was made up of the preceding words in the same sentence. In the remaining datasets (Luke & Christianson, 2018; Boyce & Levy, 2023), we included the whole preceding passage, comprising multiple sentences. For critical words made up of multiple tokens, surprisal was calculated as the sum of all the sequential tokens comprising them.

We ran regression analyses for each dataset using linear mixed-effects regression models, predicting each human language comprehension metric using the surprisal calculated using each language model, as well as baseline covariates and random effects structures as described in Appendix A. For each regression, we calculate the Akaike Information Criterion (AIC; Akaike, 1973), a measure of how well a regression fits the data, with a lower AIC indicating a better fit. All language models were run in `Python` (Van Rossum & Drake, 2009) using the `transformers` (Wolf et al., 2020) library with `pytorch` (Paszke et al., 2019), `pandas` (McKinney, 2010), and `numpy` (Harris et al., 2020); and analyses were carried out in `R` (R Core Team, 2023) using `Rstudio` (Posit team, 2023) with the `tidyverse` (Wickham et al.,

2019), lme4 (Bates et al., 2015), and scales (Wickham et al., 2023) packages. Code and data are available at https://github.com/jmichaelov/recurrent-vs-transformer-modeling.

## 4 Results

### 4.1 N400 Datasets

Scale impacts model performance at a range of tasks, so we consider the differences between models while accounting for scale. Following previous work (e.g., Oh & Schuler, 2023a), we consider differences between models when accounting for model size and for model perplexity. While the two are generally correlated, perplexity can help explain the effect of model size. Better models might align better with metrics of human language comprehension given our own powerful predictive capabilities (Monsalve et al., 2012; Goodkind & Bicknell, 2018; Michaelov et al., 2022), but by the same token, language models may learn to predict words *too well* to model human language comprehension (Oh et al., 2024).

We first consider the results arranged by model size (Figure 1A). Overall, we find that in most cases, Mamba and RWKV performance is better than that of Pythia, and Mamba is also better than RWKV. On the Federmeier et al. (2007) data, Mamba outperforms Pythia at all model sizes. On the Michaelov et al. (2024), Szewczyk & Federmeier (2022), and Wlotko & Federmeier (2012) datasets, Mamba is better at all but one scale. Lastly, on the Szewczyk et al. (2022) and Hubbard et al. (2019) datasets, Mamba is better for all but two model sizes (and roughly equal at an additional one for the latter dataset). On the Federmeier et al. (2007), Hubbard et al. (2019), and Szewczyk & Federmeier (2022) datasets, RWKV outperforms Pythia at all but one size. For the other studies, RWKV outperforms Pythia at all but 2 sizes. For all studies, the best model fit across all model sizes is a recurrent model; either Mamba or RWKV.

One additional pattern relates to scaling. In contrast to recent work on reading time (e.g. Oh & Schuler, 2023b) but in line with previous work on the N400 (Merkx & Frank, 2021;

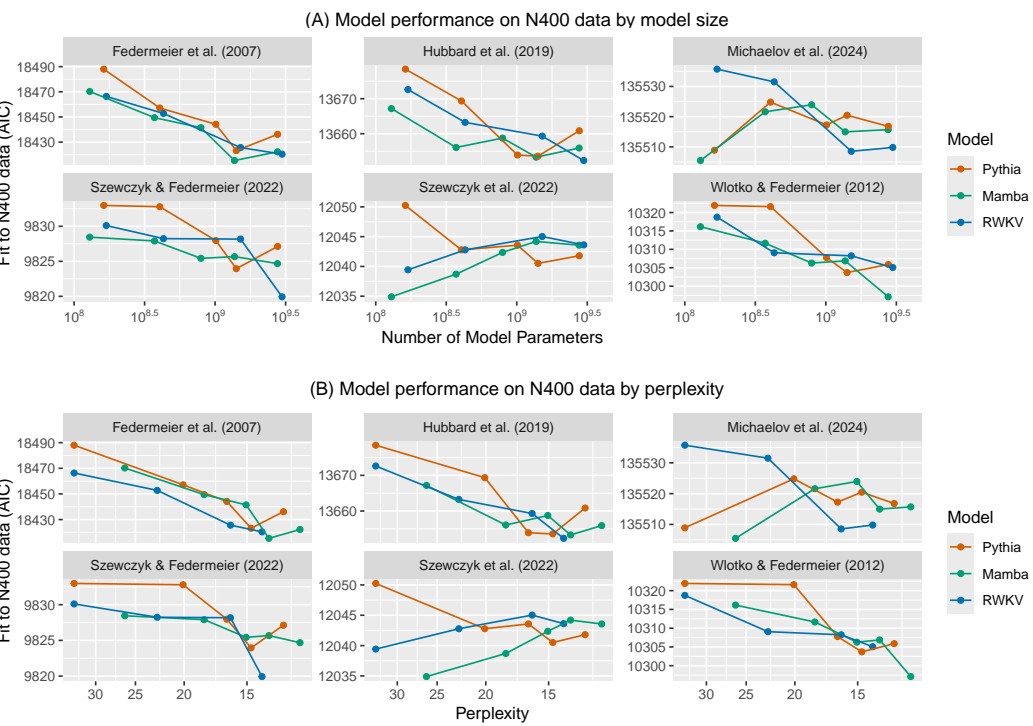

Figure 1: Language model performance at predicting N400 amplitude.

Michaelov et al., 2022; Michaelov & Bergen, 2022), we see that 4 of the 6 datasets (Federmeier et al., 2007; Hubbard et al., 2019; Szewczyk & Federmeier, 2022; Wlotko & Federmeier, 2012) show positive scaling effects—larger models tend to fit the data better.

In order to test how robust these patterns are, we run ordinary least-squares linear models for each dataset, predicting the AIC of the linear mixed-effects regressions based on language model scale and model architecture (Pythia, Mamba, or RWKV). After correction for multiple comparisons (Benjamini & Yekutieli, 2001), we see that model scale is a significant predictor of AIC, with surprisals calculated from larger models fitting the N400 data from 4 of the 6 datasets (Federmeier et al., 2007; Hubbard et al., 2019; Szewczyk & Federmeier, 2022; Wlotko & Federmeier, 2012) significantly better than smaller models. Given the low power of our analysis (only 14 observations per dataset), it is also worth noting that before correction for multiple comparisons, Mamba models produce surprisals that fit the N400 data significantly better than Pythia models on the Federmeier et al. (2007) and Wlotko & Federmeier (2012) datasets. While these latter results are suggestive rather than conclusive, they are consistent with the patterns observed in Figure 1A. The full results of our statistical analyses are provided in Table 3.

Next, we consider the results arranged by model perplexity (Figure 1B). Within each architecture, there is no difference in pattern depending on whether we order language models by size or perplexity. However, we do see a difference across architectures. In the four datasets that show positive scaling as a function of model size (larger models predict N400 amplitude better), when arranged by perplexity, Mamba models appear to perform worse relative to the other model architectures than they do when arranged by model size, while RWKV models appear to perform better. Conversely, on the dataset where two recurrent models show negative scaling (Szewczyk et al., 2022), we see the opposite pattern—Mamba appears to perform better, and RWKV appears to perform worse.

When we run ordinary least-squares linear models predicting AIC based on model perplexity and architecture, we see a similar effect to that seen for size. After correction for multiple comparisons, better language models (i.e., those with a lower perplexity) produce surprisals that better fit the N400 data on half of the datasets (Federmeier et al., 2007; Hubbard et al., 2019; Wlotko & Federmeier, 2012). The Szewczyk & Federmeier (2022) dataset also shows this pattern before correction. Full results of our statistical analyses are provided in Table 5.

## 4.2 Reading Time Datasets

For the behavioral reading data, we again first look at the data arranged by model size (Figure 2A). The clearest effects are seen on the eye-tracking datasets (Kennedy et al., 2003; Luke & Christianson, 2018), where Pythia outperforms (or in one case, performs equally as well as the better of) Mamba and RWKV at all sizes. We also see that Pythia tends to perform best overall on two of the other datasets (Boyce & Levy, 2023; Futrell et al., 2021), with either Mamba or RWKV performing better at one or two sizes. The clearest exception to this pattern is the Brothers & Kuperberg (2021) dataset, where Mamba and RWKV outperform Pythia at all but one size. We also see a different scaling pattern—unlike the the 5 datasets (Boyce & Levy, 2023; Futrell et al., 2018; Kennedy et al., 2003; Luke & Christianson, 2018; Smith & Levy, 2013) that generally show a negative scaling pattern(larger models perform produce less well-fitting surprisals), the Brothers & Kuperberg (2021) shows the positive scaling effect seen with the N400 data. The Smith & Levy (2013) results are less clear, both in terms of differences between models and in terms of overall scaling patterns.

An ordinary least-squares linear model predicting AIC based on number of parameters and architecture also shows this difference. Even after correction for multiple comparisons, model size has a significant effect on 4 of the 6 datasets (Boyce & Levy, 2023; Futrell et al., 2021; Kennedy et al., 2003; Luke & Christianson, 2018; Smith & Levy, 2013; in addition to the Smith & Levy (2013) dataset before correction for multiple comparisons), with the surprisal calculated from larger models showing a worse fit to the data. Intriguingly, in line with the aforementioned observations based on Figure 2A, before correction, the Brothers & Kuperberg (2021) dataset shows the opposite effect—the same positive scaling we see on some of the N400 datasets. Returning to the differences between architectures,

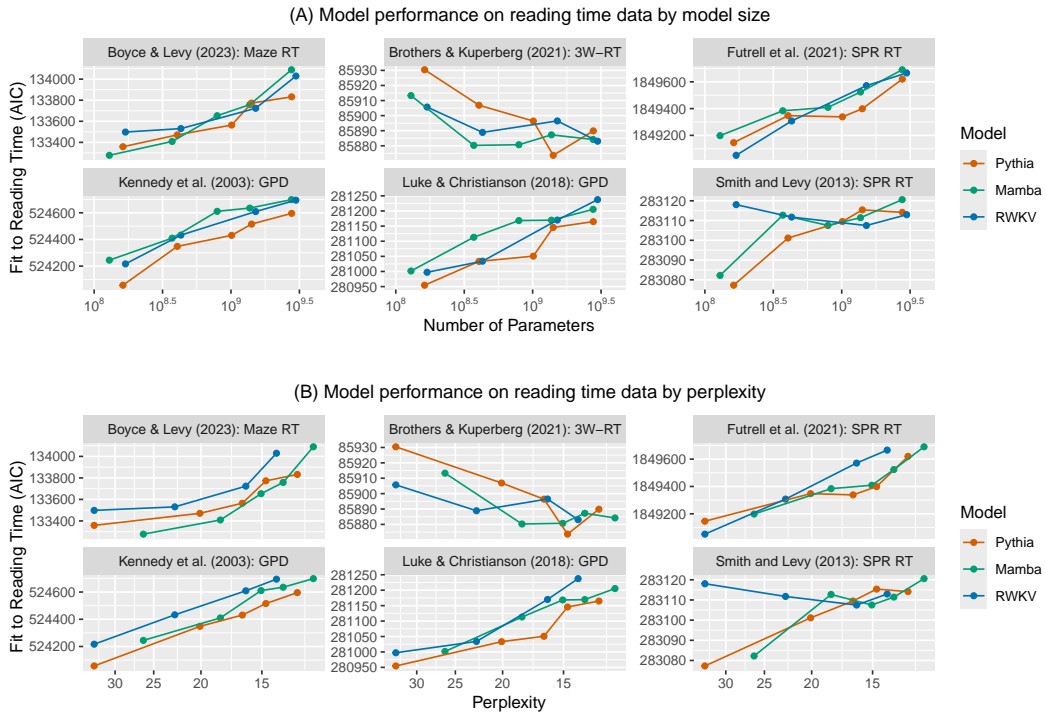

Figure 2: Language model performance at predicting 4 reading time metrics (see §3.2).

after correction, surprisals calculated using Mamba fit the Luke & Christianson (2018) and Kennedy et al. (2003) data significantly worse than Pythia, with this also being true of RWKV on both datasets before correction. Further details are provided in Table 4.

For the perplexity-ordered data (Figure 2B), the same pattern emerges as for the N400 data—for the dataset where positive scaling is found (Brothers & Kuperberg, 2021), Mamba models appear to perform relatively worse relative to Pythia than they do when the models are ordered by size and RWKV models perform relatively better, and for datasets where negative scaling is found, Mamba models appear to perform relatively better and RWKV models relatively worse. Again, while we see N400-like positive scaling on the Brothers & Kuperberg (2021) dataset—with lower-perplexity models showing a better fit to the data—we see the opposite pattern with the remaining 5.

Further confirmation of the different scaling patterns comes from ordinary least-squares linear models predicting AIC based on perplexity and architecture. After correction for multiple comparisons, models with a lower perplexity produce surprisals that are significantly better at predicting the Brothers & Kuperberg (2021) data, but significantly worse at predicting reading time in 4 datasets (Boyce & Levy, 2023; Futrell et al., 2021; Kennedy et al., 2003; Luke & Christianson, 2018; and again, the Smith & Levy, 2013 dataset before correction). In this analysis, surprisal values calculated from the Mamba and RWKV models also show a significantly worse fit to the Kennedy et al. (2003) data, with those from the RWKV models also showing this on the Luke & Christianson (2018) dataset before correction for multiple comparisons. The details of all statistical analyses are provided in Table 6.

## 5  Discussion

To the best of our knowledge, the present study is the first to compare the extent to which transformers and contemporary recurrent language models can model online human language comprehension. Previous work has overwhelmingly found that transformers better predict the N400 than recurrent neural networks (Merkx & Frank, 2021; Michaelov et al.,

2021; 2022). We show, by contrast, that on 6 datasets, when comparing models of the same size and trained on the same data, contemporary recurrent language model architectures generally out-perform transformers, with surprisal values calculated using Mamba models tending to provide the best fit to the N400 data. When accounting for model perplexity, the comparison across architectures is less clear-cut; however, the contemporary recurrent architectures at least match transformer performance.

The results are more mixed for reading time metrics. On the Kennedy et al. (2003) dataset, for example, the Pythia models predict go-past duration best at any scale or perplexity; while on the other hand, the recurrent models predict self-paced reading time on the Brothers & Kuperberg (2021) dataset best except at the 1.4-1.5B scale. Such mixed results for behavioral data is perhaps unsurprising given the conflicting results in previous work (Goodkind & Bicknell, 2018; Merkx & Frank, 2021; Hao et al., 2020; Wilcox et al., 2020; 2023a; Kuribayashi et al., 2021; Oh et al., 2022; 2024; Oh & Schuler, 2023a;b; Shain et al., 2024).

We also report several interesting scaling results. First, on the whole, scaling patterns are consistent across architectures. For datasets where larger, lower perplexity models tend to predict the human metric better (Federmeier et al., 2007; Hubbard et al., 2019; Szewczyk & Federmeier, 2022; Wlotko & Federmeier, 2012; Brothers & Kuperberg, 2021), this tends to be true for all model architectures. The same is true for datasets where smaller models and those with a higher perplexity tend to predict the human metric better (Boyce & Levy, 2023; Futrell et al., 2021; Kennedy et al., 2003; Luke & Christianson, 2018). The Szewczyk et al. (2022) and Smith & Levy (2013) datasets offer possible exceptions, where different models appear to show different scaling effects. However, without more models of each architecture, it is impossible to be certain.

Another surprising result is that contrary to the recent work that finds the same negative scaling pattern across all reading time datasets (including both self-paced reading and eye-tracking metrics; Oh & Schuler, 2023b; Oh et al., 2024), here one dataset (Brothers & Kuperberg, 2021) actually showed positive scaling. One possible explanation for this is that it is not log-transformed like the other reading time metrics. Another is that it includes the reading times of the following two words. However, the fact that it is slightly better predicted by taking the surprisal of the first word rather than that of the combined three words (see Shain et al., 2024, SI 1) suggests that the metric itself may not differ in this way as much as may be expected. A more likely explanation for the difference is that unlike the other behavioral studies which involved the reading of naturalistic stimuli, the stimuli in the Brothers & Kuperberg (2021) were carefully constructed to have different degrees of predictability. All the N400 studies use such stimuli, and this may therefore explain why the Brothers & Kuperberg (2021) results more closely resemble the positively-scaling N400 results. In any case, the finding highlights the point made by Brothers & Kuperberg (2021) that the task and stimuli used in such studies should not be overlooked when making wider claims about the relationship between probability and processing difficulty. It further suggests that the recent and ostensibly robust findings of negative scaling with behavioral data (Oh et al., 2022; 2024; Oh & Schuler, 2023a;b) may be limited to a specific type of reading time study, and that further analyses should be carried out.

Finally, we note the finding that when comparing architectures by model perplexity rather than model size, there was a consistent pattern in terms of which model best predicted the data. Specifically, compared to when ordered by model size, when the dataset showed positive scaling, the performance of Mamba appeared worse relative to other architectures, and the performance of RWKV appeared better; and when the dataset showed negative scaling, the reverse was true. Given that at each size, Mamba has a lower perplexity than Pythia and RWKV has a higher perplexity (Gu & Dao, 2023; Appendix B), this suggests that a language model's ability to predict the next word in a sequence does impact the extent to which it can model online human language comprehension above and beyond model size and architecture. Specifically, this result suggests that there are scaling effects across model architectures related to model quality (i.e., performance at next-word prediction). Even when controlling for number of parameters and training data, on a dataset that exhibits positive scaling, models that are better at next-word prediction are better at the human metric; and the converse is true for datasets that exhibit negative scaling.

### 5.1 Theoretical implications

Ultimately, the results highlight a number of complicating facts. First, there is no single universal pattern accounting for the relationship between language model probability and all metrics of online human language comprehension. Second, general language modeling performance has an effect on the extent to which language models can predict such metrics. And third, there are idiosyncratic differences between datasets, metrics, and model architectures.

Nonetheless, the present study opens up new lines of research. Crucially, in contrast to previous work, the results show that transformers are not uniquely well-suited to modeling the N400. They also align with previous research showing the same for some measures of reading time (Eisape et al., 2020; Kuribayashi et al., 2021; Merkx & Frank, 2021; Oh et al., 2022). Indeed, in our results, the differences in modeling performance between models of different architectures at a given scale or perplexity tend to be dwarfed by the differences within architectures across these dimensions.

In the present study, the performance of transformers and recurrent models is comparable, and thus our results are not able to evaluate whether there are specific architectural features of transformers or the recurrent models that make them better able to model human language comprehension. As discussed in §2, recurrent models provide a better model of the role of the working memory bottleneck (Merkx & Frank, 2021; Michaelov et al., 2021), while transformers better simulate cue-based retrieval models of comprehension (Ryu & Lewis, 2021; Merkx & Frank, 2021). It is not necessarily straightforward to disentangle the two. For example, Ryu & Lewis's argument that transformers are good models of cue-based retrieval is based on the finding that they display interference effects on agreement (also known as agreement attraction effects) and show patterns in attention that align with the theory. But recurrent neural networks have been observed to display such effects (Arehalli & Linzen, 2020), which could be plausibly explained by lossy compression of the context. Nonetheless, identifying cases where the behavior of recurrent models differs from that of transformers qualitatively—rather than quantitatively, as in the present study (with the possible exception of the Szewczyk et al., 2022 data)—and comparing these to the human data is likely to be valuable across the computational-to-cognitive model continuum. At the more purely computational end, it is important to know which type of model to use to best capture the possible effects of statistics on language comprehension. Further towards the direction of cognitive modeling, such experiments can help to evaluate which theories provide more viable explanations of human language comprehension, for example, by comparing the predictions of the now-or-never bottleneck and cue-based retrieval accounts.

The new generation of recurrent models is in its infancy. As these models continue to be developed, optimized, and scaled up, the question of whether they or transformers provide better models of human language comprehension (or at least, show a stronger degree of correlation to specific metrics of online human language comprehension) is likely to become clearer. In the meantime, the results presented here suggest that recurrent models not only match, but in some cases exceed, the performance of contemporary transformers at modeling human language comprehension, and may provide a valuable way to test hypotheses about the neurocognitive mechanisms underlying it.

## 6 Conclusions

We compare how well transformers and two contemporary recurrent language model architectures—RWKV and Mamba—can predict 5 different metrics of online human language comprehension. We find that overall, the recurrent models tend to match the performance of transformers at predicting both neural and behavioral human metrics, and that when specifically comparing across architectures by number of model paramaters, recurrent models in fact appear to be best at predicting N400 amplitude.

**Acknowledgments**

We would like to thank the San Diego Social Sciences Computing Facility Team for the use of the Social Sciences Research and Development Environment (SSRDE) cluster. Models were evaluated using hardware provided by the NVIDIA Corporation as part of an NVIDIA Academic Hardware Grant.

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

# A  Data and Analysis Details

## A.1  N400 Amplitude

The N400 is a negative-going component of the event-related brain potential that occurs roughly 300-500ms after the presentation of a stimulus, peaking at around 400ms (Kutas & Hillyard, 1980). A well-replicated finding is that the amplitude of the N400 response to a word is sensitive to the contextual probability of a word, either operationalized as cloze probability (Kutas & Hillyard, 1984)—the proportion of people to fill in a gap in a sentence with a given word (Taylor, 1953; 1957)—or when using the predictions of language models (Frank et al., 2015). Specifically, the amplitude of the N400 response elicited by a word is large by default, and decreases by the extent to which it is predictable based on the preceding context.

In this study, we compare how well the Pythia, RWKV, and Mamba models predict N400 amplitude based on the results of 6 experiments (Federmeier et al., 2007; Hubbard et al., 2019; Michaelov et al., 2024; Szewczyk & Federmeier, 2022; Szewczyk et al., 2022; Wlotko & Federmeier, 2012). The details of these datasets and how they were analyzed are outlined below.

**Federmeier et al. (2007)**  measured N400s to low- and high-cloze words in low- and high-constraint contexts. We use the data from this study as preprocessed by Szewczyk & Federmeier (2022). In this dataset, N400 amplitude is operationalized as the mean voltage at four centro-parietal electrodes (MiCe, MiPa, LMCe, RMCe) over the 300-500ms time window. N400 amplitudes are also not baseline-corrected; instead, the mean amplitude in the -100-0ms time window is intended to be included as a covariate in analysis. This dataset contains 7856 trials from 32 participants reading 564 stimuli.

To calculate model fit to the N400 data, we followed as closely as possible the approach used by Szewczyk & Federmeier (2022), which involved predicting N400 amplitude using a linear mixed-effects regression with surprisal, baseline amplitude, log-transformed frequency, the position of the word in the sentence, orthographic neighborhood distance, and concreteness as fixed effects. We also used the same random effects structure, removing variables until a structure that would not lead to singular fits for any regression was reached, which included random slopes of baseline for each subject and experimental item, random slopes of word position for each subject, and random intercepts of subject and item. We then compared the fit of regressions using surprisal calculated from each language model.

With the exception of the Michaelov et al. (2024) dataset, our remaining N400 datasets are the others provided online by Szewczyk & Federmeier (2022), and thus are preprocessed and analyzed in the same way.

**Wlotko & Federmeier (2012)**  used stimuli from Federmeier et al. (2007) as well as (Wlotko & Federmeier, 2007), which were selected to cover a wide range of probabilities. This dataset was made up of 4,440 trials (300 stimuli; 16 experimental participants).

**Hubbard et al. (2019)**  used 192 stimuli from Federmeier et al. (2007). The dataset comprises of 5,705 trials (32 participants).

**Szewczyk et al. (2022)**  also based their stimuli on those in Federmeier et al. (2007), with adjectives added before critical words, making them more or less predictable. The dataset is comprised of 4,939 trials (672 stimuli; 32 participants).

**Szewczyk & Federmeier (2022)**  also release an additional dataset with data from a previously-unpublished study using stimuli based on Federmeier et al. (2007) and including data from 4,822 trials (600 stimuli; 26 experimental participants). We refer to this as the Szewczyk & Federmeier (2022) dataset.

**Michaelov et al. (2024)**   The stimuli of the Michaelov et al. (2024) dataset differ from the other datasets in their design. Rather than having two versions of each sentence—the most likely continuation and an unlikely one—each sentence has four possible endings: the highest-cloze continuation, a low-cloze but plausible continuation that is semantically related to this highest-cloze continuation, an equally low-cloze but unrelated continuation, and an implausible continuation. The two low-cloze completions were matched for cloze probability and plausibility. There were 125 sentence frames, for a total of 500 sentences. There were fifty participants, and data from a total of 5,526 trials after cleaning.

The N400 was operationalized as the mean voltage in the 300-500ms time-window at each of the C3, Cz, C4, CP3, CPz, CP4, P3, Pz, and P4 centro-parietal electrodes. Unlike the data released by Szewczyk & Federmeier (2022), the voltage at each electrode was treated as a separate data point and N400 amplitudes were baselined using the mean amplitude in the 100ms period before stimulus presentation. Thus this dataset comprises of 49,734 data points. We analyzed these data in the same way as in Michaelov et al. (2024), fitting a regression that predicted N400 amplitude using Surprisal, log-transformed word frequency, orthographic neighborhood distance as main effects, and included random intercepts of experimental subject, sentence context, critical word, and electrode.

## A.2   Self-Paced Reading Response Time

Self-Paced Reading is an experimental paradigm in which participants read a text one word at a time, pressing a button or key to proceed to the next word. The reading time of a word is the time taken between button presses (i.e., between pressing the button to proceed to that word and pressing the button to proceed to the next word). Self-Paced Reading Response Time is generally considered to reflect processing difficulty, with longer reading times indexing a more difficult word.

**Futrell et al. (2021)**   The Natural Stories Corpus (Futrell et al., 2021) is made up of self-paced reading times from 181 experimental participants reading 10 texts, each comprising roughly 1000 words. These texts were constructed by taking publicly available texts and editing them to contain rare and hard-to-process syntactic constructions. Following Oh et al. (2024), we excluded reading times for all words that appeared at the beginning or end of a sentence, all reading times shorter than 100ms or longer than 3000ms, and all data from participants who answered three or fewer comprehension questions correctly. The regression used to predict log-transformed reading time also followed that described by Oh et al. (2024). In addition to language model surprisal surprisal, we included word length, log-transformed word frequency, and the word's position in the sentence as predictors, as well as random slopes of each of these predictors for each subject, and random intercepts of subject and sentence.

**Smith & Levy (2013)**   This dataset is comprised of self-paced reading times from 35 experimental participants reading 292-902 word passages from the Brown Corpus of American English (Francis & Kučera, 1964), for a total of 2860-4999 words per subject. We used the same exclusion criteria as with the Natural Stories data (with the exception of the comprehension score, which was not available), and the linear mixed-effects regressions each had the same structure as those used in analyzing the Natural Stories data.

## A.3   Self-Paced Reading Three-Word Response Time

A well-known phenomenon in self-paced reading is that of spillover effects, where the extent to which a word is difficult to process also impacts the reading time of one or more following words. One way to account for this is to use the reading time of a given word and the following two words as a measure of the processing difficulty associated with the word.

**Brothers & Kuperberg (2021)**   In this self-paced reading study, there were 216 sentence sets, each in a low-, medium-, or high-cloze condition, for a total of 648 stimulus sentences. Participants were excluded by Brothers & Kuperberg (2021) if they had an average comprehension check score of less than 75%. After exclusions, data from 216 of the total 240

participants were included in the analysis with a total of 46,092 data points. Data were cleaned and preprocessed by Brothers & Kuperberg (2021). We fit regressions following the method in the original study, predicting reading time (un-transformed) using a linear mixed-effects model with a main effect of language model surprisal, random intercepts for each subject and item, and random slopes of surprisal for each subject and item.

## A.4 Maze Task

Like self-paced reading, in the Maze task, participants read a text one word at a time. However, in the Maze task, participants see pairs of words and can only proceed to the next word in the text by choosing the correct next word on the screen. If the participant chooses the incorrect word, they receive feedback and are prompted to choose again. The time it takes for participants to choose a word is recorded as the reaction time. We look at the reaction times from a previous study by Boyce & Levy (2023). Dataset and analysis details are provided below.

**Boyce & Levy (2023)**    In this study, participants completed a Maze task using the stimuli from the Natural Stories corpus (Futrell et al., 2021), which comprises 10 texts based on publicly available texts, each approximately 1000 words long. In total, Natural Stories contains 10,245 words. Boyce & Levy (2023) recruited 100 participants, but participants were excluded if they did not self-report as native speakers of English. Following Boyce & Levy (2023) and Shain et al. (2024), we exclude data for all words with a reading time of less than 100ms or greater than 5000ms, incorrect words, words that were at the start or end of a sentence, and all data from participants that correctly answered fewer than 80% of comprehension questions correctly. We construct linear mixed effects regressions predicting log-transformed reaction time with surprisal, word length, log-transformed word frequency, and the word's position in the sentence. We also included random slopes of surprisal, word length, and word position for each subject, as well as a random intercept of sentence. Our analysis is based on the preprocessed version of this dataset provided by Shain et al. (2024).

## A.5 Go-Past Duration

Go-past duration is an eye-tracking-based metric of reading time. In eye-tracking studies, participants generally read a text naturalistically. Unlike in the other experimental paradigms, participants can see the whole text at one time and are able to look at previously read words. The location of each participant's gaze is recorded using an eye tracker, which also records how long participants' gaze is fixated on a given location. There are many different possible eye-tracking metrics for a given word that can be calculated (see, e.g., Shain et al., 2024), but following recent work analyzing how well different language models predict eye-tracking data (Oh & Schuler, 2023b;a; Oh et al., 2024), we look at log-transformed go-past duration, which is defined as the amount of time from when the word was first fixated to when the participant first looked to the right of that word (in left-to-right languages like English; see Luke & Christianson, 2018; Shain et al., 2024). We use data from the Provo corpus (Luke & Christianson, 2018). The details of this dataset and how it was analyzed are provided below.

**Luke & Christianson (2018)**    The Provo Corpus (Luke & Christianson, 2018) is a dataset consisting of eye-tracking data for 84 participants reading 55 passages (news articles, popular science magazines, and fiction). Passages averaged 50 words long. In total, the texts comprised 2,689 words. Participants' go-past durations were recorded while they read each text. As with the N400, we use linear mixed-effects regressions to calculate the fit of the surprisals calculated by each model to the data. Following recent work (Oh et al., 2024), we exclude from our analysis all words that were not fixated, that followed saccades of longer than 4 words, and that were at the start or end of sentences or files. Also following Oh et al. (2024), we constructed a regression to predict log-transformed go-past duration based on surprisal as well as the following covariates: saccade length (in words), word length (in characters), word position in the sentence, log-transformed word frequency, and whether the previous word was fixated. We also included random slopes of all predictors for each

subject, as well as random intercepts for each subject and sentence. Our analysis is based on the preprocessed version of this dataset provided by Shain et al. (2024) which was combined with the full set of stimuli provided by Luke & Christianson (2018).

**Kennedy et al. (2003)** The Dundee Corpus (Kennedy et al., 2003) is a dataset consisting of eye-tracking data from 10 participants reading 20 text files, each roughly 2,800 words in length, for a total of 56,212 words (Kennedy & Pynte, 2005; Kennedy et al., 2013, for additional details, see). Our analysis approach was the same as for the Provo Corpus (Luke & Christianson, 2018): exclusion criteria for data were the same (except that, following Oh et al., 2024, we also exclude words at the start and end of lines and of the screen, which are not annotated in the Provo Corpus), as was the structure of each linear mixed-effects regression. Our analysis is based on the preprocessed version of this dataset provided by Shain et al. (2024).

# B Comparison of model scale and perplexity

In the original Mamba paper, Gu & Dao (2023) report the performance of Mamba models on a number of benchmarks against comparable language models with different architectures. In the 1.4-1.5B and 2.8-3B parameter range, Gu & Dao (2023) find that Mamba has a lower perplexity than Pythia on the Pile (Gao et al., 2020) validation set, and that RWKV has a higher perplexity. This result suggests that at these scales, the Mamba models are best able to learn the statistics of language, followed by the Pythia transformers, which are in turn followed by the RWKV models.

We further replicate this finding for the other model sizes in Figure 3, finding that with the exception of the smallest models (130-170M parameters) where Pythia and RWKV have the same perplexity, at every model size, Mamba has the lowest perplexity, followed by Pythia, followed by RWKV.

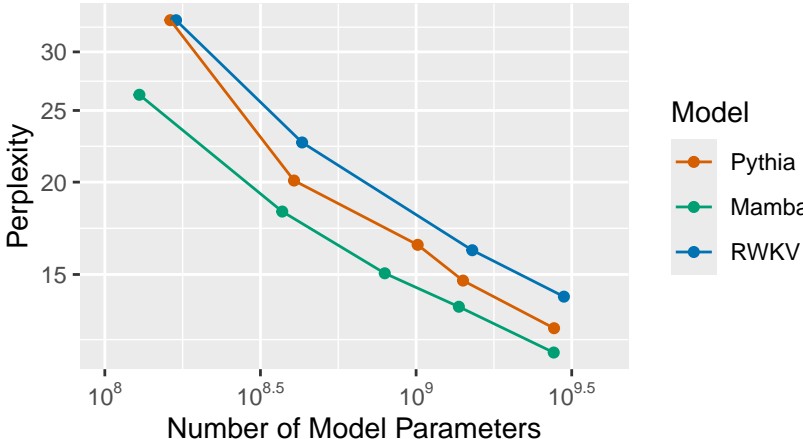

Figure 3: Comparison of the WikiText perplexity of each model of each architecture. Word-level perplexity of the WikiText-2 test set (Merity et al., 2017) was calculated using the Language Model Evaluation Harness (Gao et al., 2021).

# C Statistical Analyses

## C.1 Fit to N400 data by model size

| Dataset | Predictor | Estimate | SE | t (10) | p | p (uncor.) |
|---|---|---|---|---|---|---|
| **Federmeier et al. (2007)** | Intercept | 0.3139 | 0.1737 | 1.8068 | 1.0000 | 0.1009 |
| | *Mamba* | *-0.5605* | *0.2459* | *-2.2797* | *0.6657* | *0.0458* |
| | RWKV | -0.3979 | 0.2605 | -1.5276 | 1.0000 | 0.1576 |
| | **Scale** | **-0.9164** | **0.1078** | **-8.4972** | **0.0003** | **<0.0001** |
| **Hubbard et al. (2019)** | Intercept | 0.3005 | 0.2758 | 1.0897 | 1.0000 | 0.3014 |
| | Mamba | -0.7025 | 0.3904 | -1.7997 | 1.0000 | 0.1021 |
| | RWKV | -0.1738 | 0.4136 | -0.4202 | 1.0000 | 0.6832 |
| | **Scale** | **-0.7949** | **0.1712** | **-4.6428** | **0.0267** | **0.0009** |
| **Michaelov et al. (2024)** | Intercept | -0.0591 | 0.4811 | -0.1229 | 1.0000 | 0.9046 |
| | Mamba | -0.1731 | 0.6809 | -0.2543 | 1.0000 | 0.8044 |
| | RWKV | 0.4234 | 0.7214 | 0.5869 | 1.0000 | 0.5703 |
| | Scale | -0.2270 | 0.2987 | -0.7599 | 1.0000 | 0.4649 |
| **Szewczyk & Federmeier (2022)** | Intercept | 0.4910 | 0.2899 | 1.6940 | 1.0000 | 0.1211 |
| | Mamba | -0.8209 | 0.4103 | -2.0008 | 0.9786 | 0.0733 |
| | RWKV | -0.6925 | 0.4347 | -1.5932 | 1.0000 | 0.1422 |
| | **Scale** | **-0.7424** | **0.1800** | **-4.1257** | **0.0484** | **0.0021** |
| **Szewczyk et al. (2022)** | Intercept | 0.3879 | 0.4560 | 0.8506 | 1.0000 | 0.4149 |
| | Mamba | -0.8438 | 0.6455 | -1.3073 | 1.0000 | 0.2204 |
| | RWKV | -0.3029 | 0.6838 | -0.4429 | 1.0000 | 0.6673 |
| | Scale | 0.2281 | 0.2831 | 0.8056 | 1.0000 | 0.4392 |
| **Wlotko & Federmeier (2012)** | Intercept | 0.3373 | 0.2122 | 1.5894 | 1.0000 | 0.1431 |
| | *Mamba* | *-0.7280* | *0.3004* | *-2.4237* | *0.5711* | *0.0358* |
| | RWKV | -0.2705 | 0.3182 | -0.8501 | 1.0000 | 0.4152 |
| | **Scale** | **-0.8666** | **0.1317** | **-6.5779** | **0.0026** | **<0.0001** |

Table 3: Results of statistical analyses on the N400 datasets based on model size (operationalized as number of parameters). Because all variables were *z*-scored before analysis, the estimate does not directly reflect the difference but is helpful as an indication of effect direction—a negative estimate indicates a lower AIC, and thus, a better fit to the data. The estimate for predictors Mamba and RWKV reflects their effect relative to the Pythia models. Scale is operationalized as the logarithm of the number of parameters. We **bold** predictors that are significant after correction for multiple comparisons (Benjamini & Hochberg, 1995). Given the low power of our study (see §4), we also *italicize* variables that are significant before multiple comparisons.

## C.2 Fit to reading time data by model size

| Dataset | Predictor | Estimate | SE | t (10) | p | p (uncor.) |
|---|---|---|---|---|---|---|
| **Boyce & Levy (2023)** | Intercept | -0.2065 | 0.1726 | -1.1965 | 1.0000 | 0.2591 |
| | Mamba | 0.2558 | 0.2443 | 1.0471 | 1.0000 | 0.3197 |
| | RWKV | 0.4031 | 0.2588 | 1.5573 | 1.0000 | 0.1505 |
| | **Scale** | **0.9279** | **0.1072** | **8.6588** | **0.0003** | **<0.0001** |
| **Brothers & Kuperberg (2021)** | Intercept | 0.3774 | 0.3298 | 1.1445 | 1.0000 | 0.2791 |
| | Mamba | -0.7448 | 0.4668 | -1.5956 | 1.0000 | 0.1417 |
| | RWKV | -0.3900 | 0.4945 | -0.7887 | 1.0000 | 0.4486 |
| | *Scale* | *-0.7049* | *0.2047* | *-3.4425* | *0.1298* | *0.0063* |
| **Futrell et al. (2021)** | Intercept | -0.2114 | 0.1511 | -1.3997 | 1.0000 | 0.1918 |
| | Mamba | 0.4669 | 0.2138 | 2.1841 | 0.7605 | 0.0539 |
| | RWKV | 0.1563 | 0.2265 | 0.6902 | 1.0000 | 0.5058 |
| | **Scale** | **0.9428** | **0.0938** | **10.0537** | **0.0001** | **<0.0001** |
| **Kennedy et al. (2003)** | *Intercept* | *-0.4226* | *0.1032* | *-4.0972* | *0.0484* | *0.0022* |
| | **Mamba** | **0.7710** | **0.1460** | **5.2809** | **0.0110** | **0.0004** |
| | *RWKV* | *0.5155* | *0.1547* | *3.3326* | *0.1441* | *0.0076* |
| | **Scale** | **0.9320** | **0.0640** | **14.5533** | **<0.0001** | **<0.0001** |
| **Luke & Christianson (2018)** | *Intercept* | *-0.4129* | *0.1328* | *-3.1089* | *0.1999* | *0.0111* |
| | **Mamba** | **0.7922** | **0.1880** | **4.2141** | **0.0442** | **0.0018** |
| | *RWKV* | *0.4550* | *0.1992* | *2.2845* | *0.6657* | *0.0454* |
| | **Scale** | **0.9165** | **0.0825** | **11.1145** | **<0.0001** | **<0.0001** |
| **Smith & Levy (2013)** | Intercept | -0.3274 | 0.3627 | -0.9025 | 1.0000 | 0.3880 |
| | Mamba | 0.3384 | 0.5134 | 0.6591 | 1.0000 | 0.5247 |
| | RWKV | 0.7229 | 0.5439 | 1.3290 | 1.0000 | 0.2134 |
| | *Scale* | *0.6377* | *0.2252* | *2.8318* | *0.2931* | *0.0178* |

Table 4: Results of statistical analyses on the reading time datasets based on model size (operationalized as number of parameters). Because all variables were *z*-scored before analysis, the estimate does not directly reflect the difference but is helpful as an indication of effect direction—a negative estimate indicates a lower AIC, and thus, a better fit to the data. The estimate for predictors Mamba and RWKV reflects their effect relative to the Pythia models. Scale is operationalized as the logarithm of the number of parameters. We **bold** predictors that are significant after correction for multiple comparisons (Benjamini & Hochberg, 1995). Given the low power of our study (see §4), we also *italicize* variables that are significant before multiple comparisons.

## C.3 Fit to N400 data by model perplexity

| Dataset | Predictor | Estimate | SE | t (10) | p | p (uncor.) |
|---|---|---|---|---|---|---|
| **Federmeier et al. (2007)** | Intercept | 0.2441 | 0.1526 | 1.5991 | 1.0000 | 0.1409 |
| | Mamba | -0.1333 | 0.2184 | -0.6103 | 1.0000 | 0.5553 |
| | *RWKV* | *-0.6877* | *0.2309* | *-2.9784* | *0.2359* | *0.0138* |
| | **Perplexity** | **-0.9651** | **0.0983** | **-9.8209** | **0.0001** | **<0.0001** |
| **Hubbard et al. (2019)** | Intercept | 0.2389 | 0.2386 | 1.0013 | 1.0000 | 0.3403 |
| | Mamba | -0.3204 | 0.3413 | -0.9386 | 1.0000 | 0.3701 |
| | RWKV | -0.4356 | 0.3609 | -1.2070 | 1.0000 | 0.2552 |
| | **Perplexity** | **-0.8710** | **0.1536** | **-5.6713** | **0.0068** | **0.0002** |
| **Michaelov et al. (2024)** | Intercept | -0.0739 | 0.4882 | -0.1513 | 1.0000 | 0.8828 |
| | Mamba | -0.0934 | 0.6985 | -0.1336 | 1.0000 | 0.8963 |
| | RWKV | 0.3752 | 0.7386 | 0.5080 | 1.0000 | 0.6225 |
| | Perplexity | -0.1624 | 0.3143 | -0.5167 | 1.0000 | 0.6166 |
| **Szewczyk & Federmeier (2022)** | Intercept | 0.4354 | 0.2995 | 1.4538 | 1.0000 | 0.1766 |
| | Mamba | -0.4841 | 0.4285 | -1.1298 | 1.0000 | 0.2849 |
| | RWKV | -0.9188 | 0.4530 | -2.0281 | 0.9611 | 0.0700 |
| | *Perplexity* | *-0.7544* | *0.1928* | *-3.9127* | *0.0623* | *0.0029* |
| **Szewczyk et al. (2022)** | Intercept | 0.4018 | 0.4657 | 0.8629 | 1.0000 | 0.4084 |
| | Mamba | -0.9151 | 0.6663 | -1.3735 | 1.0000 | 0.1996 |
| | RWKV | -0.2625 | 0.7045 | -0.3726 | 1.0000 | 0.7172 |
| | Perplexity | 0.1371 | 0.2998 | 0.4574 | 1.0000 | 0.6572 |
| **Wlotko & Federmeier (2012)** | Intercept | 0.2723 | 0.2288 | 1.1904 | 1.0000 | 0.2614 |
| | *Mamba* | *-0.3345* | *0.3273* | *-1.0221* | *1.0000* | *0.3308* |
| | RWKV | -0.5350 | 0.3461 | -1.5459 | 1.0000 | 0.1532 |
| | **Perplexity** | **-0.8816** | **0.1473** | **-5.9859** | **0.0051** | **0.0001** |

Table 5: Results of statistical analyses based on model perplexity. Because all variables were *z*-scored before analysis, the estimate does not directly reflect the difference but is helpful as an indication of effect direction—a negative estimate indicates a lower AIC, and thus, a better fit to the data. The estimate for predictors Mamba and RWKV reflects their effect relative to the Pythia models. Perplexity is operationalized as negative log-perplexity in order to preserve the relationship of the other variables (where negative indicates a better fit). We **bold** predictors that are significant after correction for multiple comparisons (Benjamini & Hochberg, 1995). Given the low power of our study (see §4), we also *italicize* variables that are significant before multiple comparisons.

## C.4   Fit to reading time data by model perplexity

| Dataset | Predictor | Estimate | SE | t (10) | p | p (uncor.) |
|---|---|---|---|---|---|---|
| **Boyce & Levy (2023)** | Intercept | -0.1385 | 0.2406 | -0.5754 | 1.0000 | 0.5777 |
| | Mamba | -0.1504 | 0.3443 | -0.4369 | 1.0000 | 0.6715 |
| | RWKV | 0.6726 | 0.3640 | 1.8479 | 1.0000 | 0.0944 |
| | **Perplexity** | **0.8996** | **0.1549** | **5.8072** | **0.0060** | **0.0002** |
| **Brothers & Kuperberg (2021)** | Intercept | 0.3219 | 0.2891 | 1.1134 | 1.0000 | 0.2916 |
| | Mamba | -0.3969 | 0.4136 | -0.9596 | 1.0000 | 0.3599 |
| | RWKV | -0.6304 | 0.4373 | -1.4416 | 1.0000 | 0.1800 |
| | **Perplexity** | **-0.7990** | **0.1861** | **-4.2932** | **0.0410** | **0.0016** |
| **Futrell et al. (2021)** | Intercept | -0.1406 | 0.1704 | -0.8250 | 1.0000 | 0.4286 |
| | Mamba | 0.0371 | 0.2438 | 0.1521 | 1.0000 | 0.8821 |
| | RWKV | 0.4457 | 0.2578 | 1.7290 | 1.0000 | 0.1145 |
| | **Perplexity** | **0.9643** | **0.1097** | **8.7906** | **0.0003** | **<0.0001** |
| **Kennedy et al. (2003)** | **Intercept** | **-0.3516** | **0.0518** | **-6.7941** | **0.0021** | **<0.0001** |
| | **Mamba** | **0.3359** | **0.0740** | **4.5363** | **0.0297** | **0.0011** |
| | **RWKV** | **0.8108** | **0.0783** | **10.3565** | **0.0001** | **<0.0001** |
| | **Perplexity** | **0.9833** | **0.0333** | **29.5132** | **<0.0001** | **<0.0001** |
| **Luke & Christianson (2018)** | *Intercept* | *-0.3438* | *0.143* | *-2.4040* | *0.5722* | *0.0371* |
| | Mamba | 0.3721 | 0.2046 | 1.8182 | 1.0000 | 0.0991 |
| | *RWKV* | *0.7383* | *0.2164* | *3.4126* | *0.1310* | *0.0066* |
| | **Perplexity** | **0.9441** | **0.0921** | **10.2535** | **0.0001** | **<0.0001** |
| **Smith & Levy (2013)** | Intercept | -0.2781 | 0.3479 | -0.7994 | 1.0000 | 0.4427 |
| | Mamba | 0.0336 | 0.4978 | 0.0676 | 1.0000 | 0.9475 |
| | RWKV | 0.9313 | 0.5263 | 1.7696 | 1.0000 | 0.1072 |
| | *Perplexity* | *0.6934* | *0.2240* | *3.0960* | *0.1999* | *0.0113* |

Table 6: Results of statistical analyses based on model perplexity. Because all variables were $z$-scored before analysis, the estimate does not directly reflect the difference but is helpful as an indication of effect direction—a negative estimate indicates a lower AIC, and thus, a better fit to the data. The estimate for predictors Mamba and RWKV reflects their effect relative to the Pythia models. Perplexity is operationalized as negative log-perplexity in order to preserve the relationship of the other variables (where negative indicates a better fit). We **bold** predictors that are significant after correction for multiple comparisons (Benjamini & Hochberg, 1995). Given the low power of our study (see §4), we also *italicize* variables that are significant before multiple comparisons.

