# OpenReview forum: "Revenge of the Fallen? Recurrent Models Match Transformers at Predicting Human Language Comprehension Metrics"
_colmweb.org/COLM/2024/Conference — COLM_

### Official Review · Reviewer_LzzU · 2024-04-16

**Rating:** 6
**Confidence:** 3
**Ethics Flag:** 1

**Summary:**

This paper compares performances of the state of the art recurrent
neural networks RWKX and Mamba, with that of a transformer Pythia on 9
from a human oriented perspective, through 9 datasets by use of 4
metrics of human language processing.

The method is by computing the surprisal.  The authors empirically
show that Pythia does not necessarily outperform RWKX and Mamba,
rather the contrary. The authors argue that RNNs perform better than
Transformer, from a human-oriented perspective.

**Questions To Authors:**

-Did you compare the three systems on the same data when using the
maximum number of parameters available for each system?  If results do
you think you would get?

-Did you compare the results for other data that are not oriented to
human cognition?  If you do that, what do you think you would get?

**Reasons To Accept:**

The problem setting is interesting that is to compare the capacity of RNN
against Transformer how they perform differently for human cognitive tasks.

**Reasons To Reject:**

There are two large concerns of this paper.

1.

According to Table 1, the authors compare three models under
comparable settings. The number of parameters for every experiment is
comparable by scale.  Although I understand this is one "fair" way of
setting, I wonder whether Transformer's capability is fully exploited
this way.

Transformer performs better because it is able to accomodate larger
number of parameters than RNN, that is, O(n^2) vs. O(n), respectively,
for n being the length of the sequence.  Transformer architecture
includes that of RNN.

The present result shows that if Tranformer is trained on a system
with equivalent number of parameters of RNN, then Transformer (could)
perform worse than RNN. This is trivial to me, because Transformer
uses parameters to extract relation among tokens that cannot be
captured by RNN. Therefore for Transformer to perform fully, it must
use much more parameters than RNN.

2.

The authors focus only on human cognition.

If comparing under the same setting of Table 1, the data does not have
to be be restricted to human cognition. I guess even for a general
evaluation dataset, the same comparison of Transformer vs. RNN can be
made and presumably the same conclusion could be obtained.

In other words, since the authors do not show the evaluation results
for other evaluation dataset, the value of this work is not well
highlighted.

---

> ### Author Rebuttal · Authors · 2024-05-30
>
> We thank the reviewer for their feedback. We address the main concerns below:
>
> Fairness in model scales:
>
> We agree with the reviewer that using number of parameters to measure scale may favor some model architectures over others. This is why we also include perplexity to measure scale (Figs. 1B & 2B). Perplexity gives a measure of how good each model is overall, which allows for a comparison across architectures that is not impacted by how effective models are at using parameters.
>
>
> Focus on human cognition only:
>
> We are not sure that we fully understand the reviewer’s concern about not including performance on other types of tasks given the topic of the paper (comparing the performance of models of each architecture at predicting metrics of human language comprehension) and the submission track (“Human mind, brain, philosophy, laws and LMs”), and would appreciate additional clarification the reviewer could give. In the revised paper, we will make note that in the original Mamba paper, Gu & Dao (2023; Table 3) include a comparison of the performance of Mamba models at various tasks in comparison to other models with similar numbers of parameters (including RWKV).
>
>
> Q: “Did you compare the three systems on the same data when using the maximum number of parameters available for each system? If results do you think you would get?”
>
> We used the largest Mamba models available. For the other architectures, we picked all available sizes of comparable number of parameters to the Mamba models available, which were also trained on the Pile (see response to Reviewer sgiH above for additional details). As for the largest models of the other two architectures, we would expect even better performance on the datasets that show positive scaling (for example, Michaelov et al., 2022, cited in paper, find that GPT-3 predicts N400 amplitude better than smaller transformers), and worse performance on the datasets on which we see inverse scaling.
>
>
> Q: “Did you compare the results for other data that are not oriented to human cognition? If you do that, what do you think you would get?”
>
> In the submitted version of the paper, we calculate model perplexity on WikiText-103, and find that, at each size (number of parameters), Mamba performs best (has the lowest perplexity). Except at the smallest model size when they perform the same, Pythia performs better than RWKV at this task. As noted above, comparisons on a range of tasks can be found in Gu & Dao (2023; Table 3).

---

> > ### Comment · Reviewer_LzzU · 2024-05-31
> >
> > Thank you for your rebuttal.  After reading through other reviewers' comments and all the rebuttal, I've changed my opinion and  increased my score.

---

### Official Review · Reviewer_RQJH · 2024-04-25

**Rating:** 7
**Confidence:** 4
**Ethics Flag:** 1

**Summary:**

An empirical comparison of how different LM architectures (Transformers vs. two latest-generation recurrent models, RWKV and Mamba) predict various metrics of language processing difficulty by human subjects.

The paper starts from the somewhat suprising observation that traditional RNNs have been superseded by a less cognitively plausible architecture (Transformers) not only in a range for NLP tasks but also as predictors of behavioral metrics of language comprehension by humans.
It asks whether latest-generation RNNs would, in turn, fair better than Transformers in this context.
The comparison is performed on a suite of 9 language processing datasets using models of different architectures and sizes, all trained on the same corpus.

Results show a complex picture whereby different architectures are best across different datasets and model sizes.
Also, an earlier finding is confirmed by which processing difficulty tends to be better predicted by models that have better language modeling performance (i.e. perplexity) in general.

**Questions To Authors:**

Clarity improvements:
- AIC -> spell out acronym
- For clarity, I'd give an acronym/ID to each dataset in Table 2 and use those acronyms/IDs for the discussion in Section 4. All those citations make it hard to follow the text
- Sect. 4.1 could end with some cleared take-aways. Overall, what have you learned from this experiment?

Typos:
- identifies filters the input
- initialized in based
- In thus study
- again see -> we again see (sect 4.2)

**Reasons To Accept:**

- timely and thorough evaluation of post-Transformer recurrent architectures on the task of predicting human processing difficulty
- no less than 9 datasets and four different metrics of processing difficulty are included in the study
- models are compared across different sizes to give a more complete picture

**Reasons To Reject:**

- The results themselves are rather inconsistent and the overall conclusions a bit vague
- No attempt is made at explaining why some architectures may be outperforming others in different conditions

---

> ### Author Rebuttal · Authors · 2024-05-30
>
> We thank the reviewer for their feedback. We address the main concerns below:
>
> Inconsistent results:
>
> We agree that the results are not consistent across all datasets.
>
>
> Limitations in Discussion and Conclusion:
>
> We will substantially expand our discussion section in our revision, in which we will argue that:
>
> (1) Overall performance at predicting metrics of human language processing is similar across architectures.
>
> (2) There is a general tendency of inverse scaling between model size (or perplexity) and performance at predicting reading time, but positive scaling between model size (or perplexity) and N400 amplitude.
>
> (3) These scaling patterns are the same within architectures.
>
> (4) Some of the differences between architectures on specific datasets may be partly accounted for by general model performance—Mamba models have the lowest perplexity (i.e. are best at the pretraining task of autoregressive language modeling), and tend to perform relatively better on positive scaling datasets and relatively worse on inverse scaling datasets when considering how model parameter count impacts performance (see Figs 1A and 2A). Crucially, this pattern disappears when we focus on how model perplexity impacts performance (see Figs 1B and 2B).

---

> > ### Comment · Reviewer_RQJH · 2024-05-31
> > **thanks for the clarifications**
> >
> > Thanks for your answers. I stand by my overall positive judgement of this paper. Even in the lack of a clear winner, the paper has enough contributions with its analysis of novel models on this task.

---

### Official Review · Reviewer_WYrb · 2024-05-09

**Rating:** 8
**Confidence:** 5
**Ethics Flag:** 1

**Summary:**

The paper evaluates a recent crop of recurrent language models on predicting the N400 response and reading times, finding that the modern recurrent models perform on par with Transformers on this task, updating previous old findings that Transformers outperform LSTMs.

**Questions To Authors:**

- What is the motivation for the datasets used? Why not use Natural Stories SPR in addition to Natural Stories Maze, or any other eyetracking corpus like Dundee?

**Reasons To Accept:**

- The paper makes an important contribution to an ongoing research program about the match between neural network architectures and human language processing, based on the surprisal linking mechanism.
- The experiments are straightforward and their main result (modern recurrent architectures can match Transformers or exceed them) is clear.
- The analyses involving model size and perplexity are well-motivated and the results are interesting.

**Reasons To Reject:**

I ultimately felt dissatisfied by this paper because I wanted more, both in terms of results/experiments and in terms of cognitive upshots. The current paper feels like an incremental progress update, although the beginning of something better. I think that is probably enough for publication in CoLM, but I have concerns below:

When comparing neural architectures on their ability to predict N400/RT, the question is always what we are learning about human language processing from the comparison. I don't feel like the current paper sheds much light on this yet. What's on my mind is: What is it exactly about Transformers that made them do better than LSTMs, and what has changed about the new recurrent models that make them match Transformers? Is it the better gating mechanism in RWKV? Or is it the restriction to linear dynamics as in Mamba? I think to get real cognitive insights, more models would have to be compared, including some custom models (like, various ablated versions of Mamba). In particular, Mamba could be compared to its predecessors like S4. It would also be necessary to give LSTMs a fair shot. The LLM field basically stopped training LSTMs: how would they perform if trained on the same data and with the same large number of parameters as the modern Transformers? A comparison with n-gram models trained on large datasets would also be informative. Basically, I feel like deriving real cognitive insights would require a much more thorough comparison than what is here.

Even given the current results, the discussion about cognitive upshots is sparse. What are we to make of the relationship between scaling direction and RWKV vs. Mamba performance? If recurrent models really are just as good as Transformers, what are we to make of the arguments that Transformers act like cue-based retrieval models---was it just a spurious connection? There are rich cognitive implications that aren't explored.

Other issues

- Measuring performance by AIC is not as good as measuring performance by delta-log-likelihood on heldout data. AIC is just log likelihood with some complexity penalties. These complexity penalties would ideally be the same across different datasets, reflecting use of the same model; and if different models were used, the complexity penalties are only crude measures of the actual complexities of these models. I think pure delta-log-likelihood based on maximally similar regression models and heldout data would be the best evaluation.

- Coverage of the literature is incomplete. The most important omission in the literature review is is Ryu & Lewis (2021) who argue that Transformers directly parallel the cue-based retrieval models that are common in psycholinguistics, in contravention of the general wisdom that recurrent models are more cognitively realistic. Two other omissions are Wilcox et al. (2023, TACL) and Xu et al. (2023, EMNLP) which look at the surprisal-reading time relationship cross linguistically.

---

> ### Author Rebuttal · Authors · 2024-05-30
>
> We thank the reviewer for their feedback. We address the main concerns below:
>
> Theoretical Implications:
>
> We agree that work is needed on exactly which features of transformers and contemporary recurrent models affect performance at predicting metrics of human language processing. As the reviewer says, this work would require pre-training new (sometimes ablated) versions of Transformer and recurrent models. We view the primary value of the current study as setting the stage for this line of work. For the first time, it compares how well parameter- and training data-matched transformers, Mamba, and RWKV models can be used to predict human human metrics.
>
> We also agree that it would be valuable to expand our discussion of the theoretical stakes and implications of the study. To this end, we thank the reviewer for highlighting the cue-based retrieval account of transformers in Ryu & Lewis (2021). We also plan to expand our discussion of previous empirical work, noting that while Transformers have often been assumed to perform better than RNNs (including LSTMs) at modeling human language comprehension, this has not always been the case (Eisape et al., 2020; Merkx & Frank, 2021; and Kuribayashi et al., 2021). We also plan to expand our discussion (see response to reviewer RQJH) and suggest some theoretically-motivated avenues for future research, e.g. running analyses similar to those in Arehalli and Linzen (2020) or Ryu and Lewis (2021).
>
> Measuring performance:
>
> For each dataset, all regressions have the same structure, matching previous work as closely as possible. Given this (and the differences between the metrics), we do not believe that using the regressions with the same structure across all datasets is appropriate. However, we welcome the reviewer’s suggestions about whether calculating (R)MSE or delta-log-likelihood on a test set would nonetheless be informative.
>
> Previous work:
>
> We thank the reviewer for highlighting these studies, which we will add.
>
> Dataset Selection:
>
> Our dataset selection goal was broad coverage of experimental paradigms and metrics—in addition to the N400 studies, we included one eye-tracking, one self-paced reading, and one Maze study. In the revised manuscript, we will follow the reviewer’s suggestion to expand our analyses, adding the Dundee, the Natural Stories SPR, and Smith & Levy (2013) datasets. Having now run the analyses, these datasets show the same general pattern as the Boyce & Levy (2023) and Provo Corpus datasets.

---

> > ### Comment · Reviewer_WYrb · 2024-06-01
> >
> > Thanks for the response, I've upped my rating.

---

### Official Review · Reviewer_sgiH · 2024-05-13

**Rating:** 4
**Confidence:** 3
**Ethics Flag:** 1

**Summary:**

This is a well written paper with a clear structure. The paper is well motivated with clear arguments. The paper also presents a good amount of empirical evidence to back the conclusions.

The paper is quite original as it shows evidence that RNNs can outperform transformers in human language compression. However, in terms of technicality, the paper doesn't proposal noticeable novelty.

**Reasons To Accept:**

I enjoyed reading the paper. The paper presents well the background and the literature.

**Reasons To Reject:**

I'm skeptical about some points in the paper. Firstly, the authors argue that one reason why transformers are less cognitively plausible is their finite context window. I believe that the finite context window is only a limitation in practice because in theory, one can choose the window size as large as possible. Besides, isn't that human memory has some certain limitation?

My second concern is the models chosen in the experiments. The authors specifically chose Mamba and RWKV because they have been shown to be very fast and perform on par with transformers. However, there are several variations of transformers yet the authors didn't give a good argument why Pythia was picked.

Another related concern is about how the chosen models were trained. The paper seems lack that details. Besides, the authors draw conclusions without disentangling model architecture and model optimization. In fact, with reasonable optimization (e.g. some simple tricks), one can improve a model performance.

---

> ### Author Rebuttal · Authors · 2024-05-30
>
> We thank the reviewer for their feedback. We address the main concerns below:
>
>
> Cognitive Plausibility and Context Windows:
>
> We thank the reviewer for noting this, and agree that limited context lengths are only an issue in practice (and one that is likely to be reduced/overcome as context windows continue to increase). We also agree that human memory does have limitations—we currently discuss what kinds of limitations exist and what this means for the recurrent model vs. transformer comparison in paragraph 2 of page 3. We will expand upon both points in the revised paper.
>
>
> Model Selection:
>
> We explain model selection in the final sentence of Section 1 and the first paragraph of Section 3: all models are trained on the same dataset (The Pile), and there are 4 models of each architecture that have roughly the same number of parameters, which we show in Table 1. This allows us to compare models of different architectures while keeping the dataset constant and accounting for differences in number of model parameters, which is the main aim of the paper. We thank the reviewer for identifying this as something that could be made clearer, and will emphasize this further in the revised manuscript.
>
> Currently there are no other comparable publicly available models (trained on exactly the same dataset and of the same sizes) that are publicly available apart from the GPT-Neo models. The Pythia models are a more recent and more-optimized version of the GPT-Neo models, and so we chose them to give transformers the best chance in the comparison across architectures.
>
>
> Architecture vs. Optimization:
>
> Our aim in this study was to compare the performance of models across architectures, and (apart from GPT-Neo) there are no other comparable models of any of the three architectures. For this reason, we focused in Section 3 on the main differences between the architectures; however, if the reviewer believes that there are specific additional details that would be important to include in the paper, we would appreciate any clarification. We will mention in the revised paper that the full details of all models and their training are provided in the original papers and the linked Github repositories.
>
> Finally, we are not aware of any simple tricks that one can use to improve a model’s performance at modeling human language comprehension, and would be grateful for any more information that the reviewer could provide on this.

---

> > ### Comment · Reviewer_sgiH · 2024-06-05
> > **response**
> >
> > I would like to thank the authors for the response. My response is below:
> >
> > First, re model selection & optimization, one specific point unclear to me is how the models were trained. I couldn't find training details in the paper, and thus I can't tell if there're tricks to optimize the models.
> >
> > Second, the authors shape the contents under human cognition theme; and from cognitive plausibility point of view, the authors discuss transformers vs recurrent models. But I really don't think that's relevant because the models have several technical details that aren't proved to be cognitively plausible. How do we know that the used recurrent models outperform the transformer because their recurrent mechanism or because of their cognitively unplausible technical details?
> >
> > Third, it's unclear what the authors want to conclude here. The authors show performance results, but then what could we learn from them to help us understand human cognition?

---

> > > ### Comment · Reviewer_WYrb · 2024-06-05
> > > **Another reviewer jumps in**
> > >
> > > Jumping into this discussion as another reviewer, responding to this point:
> > >
> > > > But I really don't think that's relevant because the models have several technical details that aren't proved to be cognitively plausible.
> > >
> > > For what it's worth, as a cognitive scientist, I'd like to point out that there's already a large literature on using models to predict EEG data exactly like in this paper. It's true that some of the mechanisms in the models are cognitively implausible (like multi-head attention in Transformers), but this paper is responding to and joining existing work which has already dealt with a lot of the subtleties in linking LM surprisals to human data.

---

> > > ### Author Response · Authors · 2024-06-07
> > >
> > > We would like to thank Reviewer sgiH for their response.
> > >
> > > On the first point (optimization/training details), because we used pre-trained models, we did not include the training details and hyperparameters described in the original papers associated with each model type (Pythia: [Biderman et al., 2023](https://proceedings.mlr.press/v202/biderman23a.html), p. 2 and p. 22; RWKV: [Peng et al., 2023](https://aclanthology.org/2023.findings-emnlp.936/), pp. 14065-14066; Mamba: [Gu and Dao, 2023](https://arxiv.org/abs/2312.00752), pp. 29-36). In addition, the Pythia GitHub repository (https://github.com/EleutherAI/pythia) and the RWKV repository (https://github.com/BlinkDL/RWKV-LM) each provide the training code, and the Mamba repository (https://github.com/state-spaces/mamba) also provides code that can be used to train the models.
> > >
> > > On the second point (cognitive plausibility), we agree with the reviewer that there are certain features of the models’ implementations that may not necessarily be cognitively plausible. However, as Reviewer WYrb notes in their reply to the response, there is still value in (and an active research community engaged in) investigating the differences in how well language models of different architectures can model human language processing. In this case, as we note in the original submission and Reviewer WYrb mentions in their initial review, there are competing accounts of human language processing that can be considered analogous at a computational level to recurrent and transformer architectures—the now-or-never bottleneck in the case of recurrent models (see [Merkx and Frank, 2021](https://aclanthology.org/2021.cmcl-1.2/)), and cue-based retrieval in the case of transformers (see [Ryu and Lewis, 2021](https://aclanthology.org/2021.cmcl-1.6/);  [Merkx and Frank, 2021](https://aclanthology.org/2021.cmcl-1.2/)). We thank the reviewer for pointing out that this may not necessarily be clear in the submitted version; we will clarify and expand upon this point in the revised version.
> > >
> > > On the third point (conclusions), one of the key questions for our study was how well models of different architectures perform, for the reasons noted in the previous paragraph. Work by [Merkx and Frank (2021)](https://aclanthology.org/2021.cmcl-1.2/) suggests that transformers are indeed empirically better at predicting the amplitude of the N400 at least, and [Ryu and Lewis (2021)](https://aclanthology.org/2021.cmcl-1.6/) provide empirical and theoretical support for the idea that transformers are good models of human language processing. However, our results show that if there is a difference, it is small (and in fact, in the case of the N400, recurrent models tend to be better), which suggests that the transformer architecture may not be as important as previously thought for modeling human language processing. Instead, our results suggest that the question of which type of architecture provides a better computational-level model of human language processing is far from resolved, and that, since the new generation of recurrent models is only in its infancy, this question may need to be revisited as new developments in the field arise.

---

### Decision · Program_Chairs · 2024-07-10

**Decision:**

Accept

**Comment:**

This paper evaluates two recent recurrent models, in comparison with a transformer, on predicting measures of human language comprehension. Results suggest that -- for the specific models (with the same training data domain and comparable parameter counts), recurrent architectures show similar performance to transformers, sometimes exceeding them.

Despite discussion, this remained a controversial paper, with ratings ranging from 4 to 8. Having a strong background in this field, I believe that the weaknesses raised by sgiH (who provided the only rejection score) are satisfactorily addressed by the authors, even though sgiH did not raise their score.

Taken all reviews and my reading of the paper into consideration, I believe this to be a timely and worthwhile contribution to the sub-field evaluating language models as predictors of human language comprehension measurement.

[At least one review was discounted during the decision process due to quality]